# High Pretreatment Serum PD-L1 Levels Are Associated with Muscle Invasion and Shorter Survival in Upper Tract Urothelial Carcinoma

**DOI:** 10.3390/biomedicines10102560

**Published:** 2022-10-13

**Authors:** Ádám Széles, Petra Terézia Kovács, Anita Csizmarik, Melinda Váradi, Péter Riesz, Tamás Fazekas, Szilárd Váncsa, Péter Hegyi, Csilla Oláh, Stephan Tschirdewahn, Christopher Darr, Ulrich Krafft, Viktor Grünwald, Boris Hadaschik, Orsolya Horváth, Péter Nyirády, Tibor Szarvas

**Affiliations:** 1Department of Urology, Semmelweis University, Ulloi ut 78/b, 1082 Budapest, Hungary; 2Centre for Translational Medicine, Ulloi ut 26, 1085 Budapest, Hungary; 3Institute for Translational Medicine, Medical School, University of Pécs, Szigeti út 12., 7624 Pécs, Hungary; 4Division of Pancreatic Diseases, Heart and Vascular Center, Semmelweis University, 1085 Budapest, Hungary; 5Department of Urology, University of Duisburg-Essen, German Cancer Consortium (DTKK)-University Hospital Essen, Hufelandstraße 55, D-45147 Essen, Germany; 6Department of Genitorurinary Medical Oncology and Clinical Pharmacology, National Institute of Oncology, Ráth György utca 7-9., 1122 Budapest, Hungary

**Keywords:** upper tract urothelial carcinoma, UTUC, sPD-L1, soluble programmed death ligand-1, biomarker, prognosis, immune checkpoint inhibitor therapy, chemotherapy, radical nephroureterectomy

## Abstract

**Simple Summary:**

This study aimed to assess the prognostic relevance of soluble serum PD-L1 in upper tract urothelial carcinoma (UTUC) patients who underwent surgical or systemic (chemo- or immune checkpoint inhibitor) therapy. We found that high preoperative sPD-L1 levels were correlated with higher pathological tumor stage, grade and the presence of metastasis. In addition, higher pretreatment serum PD-L1 concentrations were associated with shorter survival in both surgically and chemotherapy-treated UTUC patients. We detected a characteristic increase in serum PD-L1 levels in UTUC patients after 3 months of anti-PD-L1 therapy. Based on these results sPD-L1 is a promising prognostic biomarker in UTUC.

**Abstract:**

Programmed death ligand-1 (PD-L1) is an immune checkpoint molecule and a widely used therapeutic target in urothelial cancer. Its circulating, soluble levels (sPD-L1) were recently suggested to be associated with the presence and prognosis of various malignancies but have not yet been investigated in upper tract urothelial carcinoma (UTUC). In this study, we assessed sPD-L1 levels in 97 prospectively collected serum samples from 61 UTUC patients who underwent radical nephroureterectomy (RNU), chemotherapy (CTX), or immune checkpoint inhibitor (ICI) therapy. In addition to pretreatment samples, postoperative and on-treatment sPD-L1 levels were determined in some patients by using ELISA. In the RNU group, elevated preoperative sPD-L1 was associated with a higher tumor grade (*p* = 0.019), stage (*p* < 0.001) and the presence of metastasis (*p* = 0.002). High sPD-L1 levels were significantly associated with worse survival in both the RNU and CTX cohorts. sPD-L1 levels were significantly elevated in postoperative samples (*p* = 0.011), while they remained unchanged during CTX. Interestingly, ICI treatment caused a strong, 25-fold increase in sPD-L1 (*p* < 0.001). Our results suggest that elevated preoperative sPD-L1 level is a predictor of higher pathological tumor stage and worse survival in UTUC, which therefore may help to optimize therapeutic decision-making. The observed characteristic sPD-L1 flare during immune checkpoint inhibitor therapy may have clinical significance.

## 1. Introduction

Urothelial carcinoma is most commonly localized in the urinary bladder; however, it can occur in every part of the urinary tract. Approximately 5–10% of all urothelial carcinomas are located in the upper urinary tract, including the ureters and pelvicalyceal system [1]. Pelvicalyceal localization of upper tract urothelial carcinoma (UTUC) is twice as frequent as the ureteral form [2]. The incidence of UTUC is approximately 1–2 cases per 100,000 each year and the 5-year disease-free survival rate ranges between 40–90% [3]. Considering their similar etiologies and therapeutic sensitivities, UTUC and urothelial bladder cancer (UBC) were considered the same disease in different anatomic locations. However, in recent years, a growing body of evidence has revealed disparities between UTUC and UBC [4]. Therefore, UTUC and UBC are now considered different tumor entities with substantial similarities. The gold standard treatment for clinically localized UTUC is surgery. In some rather rare and low-risk cases, when the biopsy confirms a small low-grade and localized tumor, endoluminal surgery or segmental resection of the ureter can be performed [5]. In the majority of cases, however, radical nephroureterectomy (RNU) with or without lymph node dissection remains the standard of care [6]. Due to the difficult anatomical features and location of UTUC, biopsy only provides a limited value for the evaluation of tumor stage [7]. As a consequence, some RNUs are performed unnecessarily, which represents an overtreatment. Therefore, preoperatively available prognostic markers are needed to reduce overtreatment of UTUC and to stratify patients for neoadjuvant systemic therapy.

In advanced cases of UTUC, systemic treatment with platinum-based chemotherapy (CTX) is the first choice in either adjuvant or neoadjuvant settings [5]. Platinum-based CTX increases both disease-free and metastasis-free survival by ~50% compared to surveillance [8]. However, after surgical removal of the kidney, the rate of cisplatin-eligible patients after RNU is only 20% [9]. For those patients who are ineligible for cisplatin, immune checkpoint inhibitors (ICI) can be administered [10]. The IMvigor 210 trial assessed the benefit of ICI therapy in cisplatin-ineligible urothelial cancer patients and revealed a never-before-seen improvement of 39% in the overall response rate [11]. However, UTUC patients show large individual differences in their responses to both CTX and ICI therapies. Therefore, pretreatment prognostication is a great unmet clinical need in UTUC.

Programmed-death ligand-1 (PD-L1) is a co-inhibitory membrane-bound protein expressed mainly by hematopoietic cells such as T-cells, B-cells, dendritic cells, and macrophages, but it is also produced by non-hematopoietic normal cells. Its receptor, programmed death protein-1 (PD-1), is primarily expressed by activated T-cells. PD-L1 is overexpressed in several cancer types. The binding of PD-1 and PD-L1 causes immune suppression, allowing tumor cells to escape from the cytotoxic effect of CD8 positive T-cells [12]. In UTUC, higher PD-L1 tissue expression levels were shown to be associated with a higher pathological tumor stage, poor survival and better response to ICI therapy [13,14,15]. Matrix metalloproteases are able to cleave the extracellular domain of PD-L1, leading to the appearance of soluble PD-L1 (sPD-L1) in the serum [16]. While sPD-L1 has recently been found to be prognostic in various malignancies, its potential clinical value in UTUC has not yet been assessed [17]. In this study, we aimed to assess the prognostic value of sPD-L1 and its changes in different treatment settings of UTUC. Therefore, in a post hoc pilot study, we determined sPD-L1 levels in prospectively collected pretreatment and on-treatment serum samples of UTUC patients who underwent either surgical or systemic (CTX or ICI) treatment.

## 2. Materials and Methods

### 2.1. Patient Cohort

Pretreatment serum samples were collected from an overall number of 61 UTUC patients (44 males, 17 females) who underwent surgical (RNU cohort; *n* = 37), postoperative platinum (CTX cohort; *n* = 25), or second-line immunotherapy (ICI cohort; *n* = 6) at the Department of Urology at Semmelweis University between August 2014 and July 2020. Six patients were included in more than one cohort (three patients were in both the RNU and CTX cohorts, two patients in the CTX and ICI cohorts, while one patient was in all three treatment groups). In addition to pretreatment samples, we collected samples following the start of therapy at predefined time points. For 14 patients of the RNU cohort, serum samples from the first postoperative day were available. Eighteen samples from the CTX cohort from the first day of the second CTX cycle, and four samples from the ICI cohort after three months of therapy were available for analysis.

Blood samples were collected in 9 mL tubes (Vacuette^®^, Greiner, Bio-One, Mosonmagyaróvár, Hungary) and left at room temperature for 30–90 min, then centrifuged with an Eppendorf 5702R centrifuge at 1500× *g* for 10 min, and finally aliquoted and kept at −80 °C until further analysis. The primary endpoint of this study was overall survival (OS), which was calculated as the period between initiation of therapy (RNU, CTX, or ICI) and the last follow-up (January 2022) or death. The secondary endpoint was progression-free survival (PFS). The study was conducted in accordance with the Declaration of Helsinki and was approved by the institutional ethics committee (TUKEB 256/2014). All patients provided a written informed consent to participate in this study.

### 2.2. Serum PD-L1 and MMP-7 Analyses

Quantitative sPD-L1 analyses were performed by using the sandwich ELISA method (PD-L1/B7-H1 Quantikine ELISA kit, DB7H10, R&D Systems, Wiesbaden, Germany), according to the manufacturer’s instructions. To exclude possible interference between the therapeutic anti-PD-L1 antibody and the used ELISA assay, we also analyzed atezolizumab (anti-PD-L1) and pembrolizumab (anti-PD-1) on our ELISA plates by adding these substances as samples to the plate. Furthermore, we added different concentrations of atezolizumab and pembrolizumab to serum samples with low and high levels of sPD-L1 to check whether the addition of therapeutic antibodies would increase sPD-L1 signals.

Serum MMP-7 levels were formerly measured by using the Human Total MMP-7 Quantikine ELISA kit (R&D Systems, Wiesbaden, Germany, Catalog Number: DMP700), according to the product instructions. In this study, MMP-7 concentrations were used for testing for a possible correlation between sPD-L1 and MMP-7. Detailed results of the MMP-7 analysis were provided in an earlier published study [18].

### 2.3. Statistical Analysis

The non-parametric two-sided Wilcoxon rank-sum test (Mann–Whitney test) was used for group comparisons. Univariate OS and PFS analyses were performed using the Kaplan–Meier log-rank test and univariate Cox analysis. Low event numbers in each cohort did not allow the performance of multivariate analyses. Receiver operating characteristics (ROC) curves were applied for the RNU and CTX treatment groups to determine sPD-L1 cut-off values with the highest sensitivity and specificity for the dichotomized endpoint of death during the follow-up period. Spearman’s rank correlation analysis was used to test for the correlation between formerly determined serum MMP-7 and sPD-L1 levels [18]. A *p*-value of < 0.05 was considered significant. All statistical analyses were performed with the IBM SPSS Statistics software (v. 27.0; IBM Corp., Armonk, NY, USA).

## 3. Results

### 3.1. Clinical Background

The median age in the RNU, CTX and ICI cohorts was 69, 72 and 65 years and the median follow-up times were 24, 18 and 20 months, respectively. In three patients, histological evaluation after RNU revealed a pT0 stage. Further patients’ characteristics and baseline sPD-L1 levels are given in Table 1.

### 3.2. Correlations of PD-L1 Concentrations with Clinicopathological Parameters

We observed no differences in the pretreatment levels of sPD-L1 between the RNU vs. CTV, RNU vs. ICI and CTX vs. ICI cohorts (*p* = 0.203, *p* = 0.391, *p* = 0.698, respectively). For the RNU cohort, age, sex, ECOG performance status and tumor localization showed no significant association with preoperative sPD-L1 levels. Higher sPD-L1 levels were found in muscle-invasive, high grade (G3) cases as well as in lymph node and/or distant metastatic cases (*p* < 0.001, *p* = 0.019 and *p* = 0.002 respectively) (Table 1, Figure 1).

For the CTX cohort, similar to the findings in the RNU cohort, baseline sPD-L1 levels were higher in metastatic cases (*p* < 0.001) (Table 1). Furthermore, patients who received gemcitabine/carboplatin instead of gemcitabine/cisplatin had significantly elevated sPD-L1 levels (*p* = 0.013). As low case numbers of the ICI cohort did not allow the performance of a valid statistical evaluation, we provided patients’ characteristics on an individual patient level in Table 2.

### 3.3. Correlation of Pretreatment sPD-L1 Levels with Patients’ Prognosis

For the RNU cohort, three patients with pT0 histopathological findings were excluded from survival analyses. Muscle-invasive disease (≥pT2) and the presence of lymphatic or distant metastases at RNU were associated with shorter OS (HR: 7.115; 95% CI 1.504–33.659; *p* = 0.013 and HR: 4.891; 95% CI 1.379–17.345; *p* = 0.014, respectively). Similarly, shorter PFS was significantly associated with the same factors: ≥pT2 stage (HR: 10.836; 95% CI 2.865–40.978; *p* < 0.001), lymph node or distant metastasis (HR: 6.185; 95% CI 2.199–17.397; *p* = 0.001) (Appendix A).

In addition, high sPD-L1 levels were associated with shorter OS, both when using the median (84.0 pg/mL; *p* = 0.041) or the ROC-based cut-off value (118.5 pg/mL; *p* < 0.001) (Figure 2A,B) (Appendix A).

For the CTX cohort, the presence of lymph node or distant metastasis at CTX baseline was found to be a significant predictor of shorter OS (HR: 14.737; 95% CI 1.810–119.987; *p* = 0.012) (Appendix A). The median sPD-L1 cut-off value for the CTX cohort was 96.1 pg/mL, which is close to the ROC cut-off of 93.9 pg/mL and therefore both cut-offs divided the CTX cohort into the very same groups (Appendix A). High sPD-L1 levels were significantly associated with shorter OS (HR: 6.956; 95% CI 1.461–33.110; *p*= 0.015) (Table 3, Figure 2). In this cohort, shorter PFS was only associated with the presence of lymph node or distant metastasis at CTX baseline (HR: 7.638; CI 95% 2.218–26.301; *p* = 0.001) (Appendix A). Survival analysis for the ICI cohort could not be performed because of low patient numbers.

### 3.4. Changes in sPD-L1 Levels during and after Therapy

In the RNU cohort, the median preoperative sPD-L1 concentration was 84.0 pg/mL. In 14 cases, postoperative (first day after RNU) sPD-L1 levels were available, with a median of 114.5 pg/mL, which was significantly higher than the pretreatment serum concentrations (*p* = 0.011) (Figure 3A,D).

In the CTX cohort, the baseline median of sPD-L1 level was 96.1 pg/mL which remained unchanged (99.4 pg/mL, *n* = 18) after the first treatment cycle (Figure 3B,D).

Interestingly, we observed a remarkable, 25-fold increase in sPD-L1 levels after 3 months of ICI treatment from 78.3 pg/mL to 1955.5 pg/mL (*p* < 0.001) (Figure 3C,D). In addition, we measured atezolizumab and pembrolizumab directly on our assay plates and found no detectable signals. Furthermore, addition of various concentrations of atezolizumab and pembrolizumab to serum samples with low and high sPD-L1 levels did not significantly influence the detected signals. These, results show that the therapeutic antibodies do not directly interfere with the ELISA assay.

### 3.5. Correlation between sPD-L1 and sMMP-7 Levels

MMP-7 has been formerly shown to proteolytically cleave membrane-bound PD-L1 releasing sPD-L1 into the circulation and we formerly determined serum MMP-7 concentrations in a largely overlapping UTUC patient cohorts [18]. Therefore, we performed an exploratory Spearmen’s rank correlation analysis between sPD-L1 and MMP-7 in the RNU, CTX and ICI cohorts, which revealed a significant positive correlation between the concentrations of these two serum proteins in the RNU cohort (*n* = 34, rs = 0.445, *p* = 0.008) but not in the ICI cohort (*n* = 5, rs = 0.900, *p* = 0.307) and CTX cohorts (*n* = 25, rs = 0.217, *p* = 0.297) (Appendix A).

## 4. Discussion

In this pilot study, we assessed the prognostic value and changes in sPD-L1 in various treatment settings of UTUC. In the surgical treatment cohort, elevated preoperative sPD-L1 was associated with the muscle-invasive stage (pT2–pT4), higher tumor grade, the presence of metastasis and poor survival. Similarly, in the CTX cohort, higher pretreatment sPD-L1 levels were associated with shorter survival. In addition, we observed a moderate but significant sPD-L1 increase in postoperative samples of the RNU cohort and a strong 25-fold increase at 3 months of ICI treatment, while sPD-L1 levels remained unchanged during CTX treatment.

Because of inaccurate preoperative biopsy and large individual differences in therapy sensitivities, clinical decision-making in UTUC is challenging. Various studies have compared the preoperative (biopsy-based) and postoperative (definitive) stage findings in UTUC, showing both high upstaging and downstaging rates of up to 60% [19,20]. Therefore, preoperative staging often cannot provide a reliable prognostic stratification. In contrast to staging, preoperative grading was found to be more consistent with postoperative histological findings, showing 97% and 62% agreement for high and low tumor grades, respectively [21]. Accordingly, biopsy grading is considered a more reliable source for prognostication. Other known preoperative prognostic factors in UTUC are advanced age, multifocality, hydronephrosis, ECOG PS ≥ 1, tobacco consumption, and delayed surgical treatment [5]. These factors, however, have only a limited value for the prediction of the clinical behavior of UTUC. Therefore, additional molecular markers that better reflect the tumor’s biological features are needed. In recent years, many potential prognostic tissue biomarkers have been investigated in UTUC including those involved in cell differentiation (Ki67, CDCA5, PAK1, INHBA, PTP4A3), cell cycle regulation (PAK1, Bcl-xL) and antitumoral immunity such as CD204+ macrophages [22]. On the one hand, these tissue markers reflect the biological behavior of cancer and could provide insight into the pathogenesis of the disease, while on the other hand, the evaluability of these biomarkers is strongly dependent on the quality of the biopsy and has not yet been prospectively validated. Much fewer studies are available on circulating biomarkers in UTUC. Promising results have been published on the prognostic relevance of preoperative serum MMP-7, neutrophil-to-lymphocyte ratio and CRP levels; however, larger validation studies are missing and therefore these markers are to be considered as experimental [18,23,24]. Preoperative sPD-L1 levels have recently been shown to be independently associated with shorter survival in colorectal and gastric cancers [25,26]. Accordingly, in the present study, we observed poor survival rates in UTUC patients with high preoperative sPD-L1 levels. Furthermore, in our RNU cohort, high sPD-L1 levels were associated with muscle-invasive tumor stage (*p* < 0.001), high (G3) tumor grade (*p* = 0.019) and the presence of metastasis (*p* = 0.002). These findings suggest sPD-L1 as a potential biomarker for the preoperative prediction of muscle-invasive disease, which may help to decide on the extent of surgical management. In addition, patients with high pretreatment sPD-L1 levels who are at higher risk of disease progression may benefit from a more aggressive therapeutic strategy (e.g., perioperative adjuvant or neoadjuvant systemic therapy). However, our results need to be confirmed in larger patient cohorts before being applied in practice.

In high-risk locally advanced UTUC cases, neoadjuvant CTX provided 47–52% pathological objective response rates and 8–10% pathological complete response rates [27], while in the metastatic setting first-line CTX provided 35–46% overall response rates [28]. These numbers reflect a considerable heterogeneity of therapy response to CTX in UTUC patients. In cisplatin-ineligible and tissue PD-L1 positive cases, first-line ICI provided improved survival with great individual differences [29]. Our present analysis revealed a significant unfavorable prognostic effect for higher baseline sPD-L1 levels in CTX-treated patients, while the limited size of the ICI cohort did not allow the performance of statistically valid survival analyses. Therefore, we could not compare the prognostic value of sPD-L1 between CTX and ICI-treated patients, which precludes a valid conclusion regarding the therapy predictive value of sPD-L1. Recently, conflicting results have been published regarding the prognostic value of baseline sPD-L1 in ICI-treated patients with various tumors [30,31,32,33,34,35]. Increased sPD-L1 levels were associated with better outcomes in esophageal and renal cell cancer patients, while in contrast in NSCLC and melanoma, high sPD-L1 levels were associated with poor survival [30,31,34,35]. Based on these results, the ICI predictive value of sPD-L1 seems to be tumor type-dependent. In addition, the associations we found between sPD-L1 with shorter survival in both the RNU and CTX cohorts suggest that sPD-L1 is prognostic rather than therapy predictive.

The origin of circulating sPD-L1 is not fully understood. Interestingly, recent studies found no correlation between soluble and tissue PD-L1 levels, suggesting that increased tissue expression is not the primary source of sPD-L1 [30,32,33]. Some members of the matrix metalloproteinase family, such as MMP-7, -9, -10, and -13 were shown to proteolytically cleave membrane-bound PD-L1, releasing the extracellular domain of the molecule into the blood circulation [36,37]. We recently assessed MMP-7 levels in serum samples of a largely overlapping UTUC cohort and found higher MMP-7 levels to be associated with higher tumor stages and the presence of metastasis [18]. Correlation analysis between serum MMP-7 and PD-L1 levels in the same samples revealed a significant direct correlation in the treatment-naive preoperative RNU samples. This finding is in line with our former observation in UBC and confirms the involvement of MMP-7 in the proteolytic degradation of sPD-L1 at a systemic level [32]. Considering the lack of association between the tissue and serum sPD-L1 levels on the one hand and the significant correlation between serum MMP-7 and sPD-L1 on the other hand, we hypothesize that the increased sPD-L1 levels are the consequence of an enhanced proteolytic tumor milieu rather than an increased PD-L1 expression of the tumor tissue.

Comparing the pre- and postoperative sPD-L1 levels, we detected a mild but significant increase after RNU, which suggests that tumor cells are not the predominant sources of sPD-L1. After surgery, patients have an elevated level of circulating damage-associated molecular patterns, which triggers a local and systemic inflammation [38]. As sPD-L1 levels were shown to be associated with the presence of severe inflammation, we hypothesize that the postoperative sPD-L1 elevation we observed may be a consequence of an inflammatory response to the surgical procedure itself [39].

In the CTX cohort, no differences were detected between the baseline and on-treatment sPD-L1 levels. In contrast, we found strong, 25-fold elevated sPD-L1 concentrations after three months of ICI treatment. These striking results are in line with our former observation made in UBC, showing a similar increase in sPD-L1 levels in PD-L1 inhibitor-treated (atezolizumab) patients after three months of therapy [32]. Interestingly, in PD-1 inhibitor-treated (pembrolizumab) UBC patients no sPD-L1 increase could be detected, suggesting that the detected sPD-L1 flare may be therapy-specific. Similar observations were described for PD-1 and PD-L1 inhibitor-treated NSCLC patients with strong sPD-L1 elevation in some cases, while in tyrosine kinase inhibitor-treated NSCLC cases, no such increase could be observed [40]. In addition, Chiarucci et al., using the same ELISA assay, found a similar increase in sPD-L1 levels (from a median of 70 pg/mL to 1850 pg/mL) in anti-PD-L1 treated mesothelioma patients, while in the anti-CTLA-4 and anti-PD-1 treated patients no such increase could be observed [41]. Based on these findings, it appears that only PD-L1 but not PD-1 inhibitors provoke an sPD-L1 increase; however, the exact mechanism of this sPD-L1 flare-up remains to be elucidated. A possible explanation for this phenomenon might be the presence of various sPD-L1 antibody complexes. Atezolizumab as an antigenic molecule may induce a significant anti-drug antibody response and the presence of this antibody might have a neutralizing effect on atezolizumab [42]. Our study has some limitations inherent to its retrospective nature and limited cohort sizes, which did not allow us to perform multivariate analyses. This limitation, however, should be judged in relation to the low incidence of UTUC. A further limitation of our study is that tumor samples were not available for correlation analysis between tissue and serum PD-L1 levels. However, data based on a large number of cases in various cancers uniformly showed no correlation between serum and tissue PD-L1 levels [30,33]

## 5. Conclusions

Assessing sPD-L1 for the first time in UTUC, we found significantly increased levels in advanced tumor stages and high pretreatment concentrations were associated with shorter survival in both RNU and CTX-treated patients. These findings, when confirmed in larger studies, may help to improve risk-stratification and to optimize therapeutic decision-making in UTUC. The characteristic sPD-L1 flare-up observed in UTUC seems to be therapy-specific and its biological and clinical relevance needs to be further evaluated.

## Figures and Tables

**Figure 1 biomedicines-10-02560-f001:**
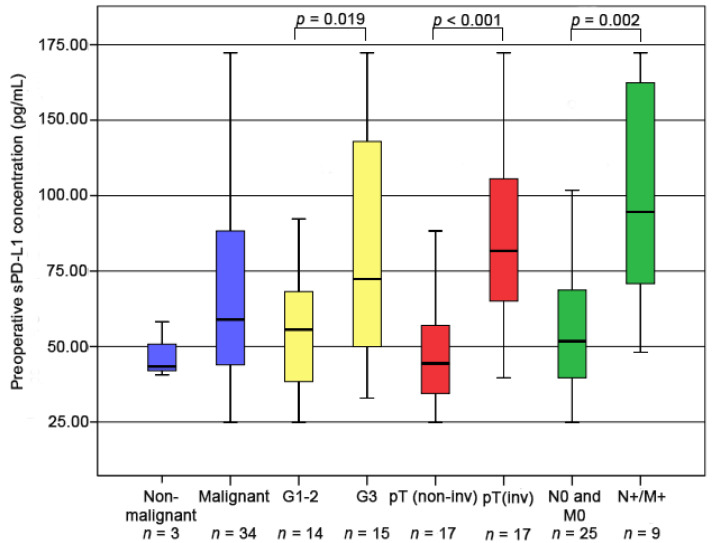
Association of preoperative sPD-L1 concentrations and clinicopathological parameters in the RNU cohort. pT(non-inv): pTa-pT1, pT(inv): pT2–pT4.

**Figure 2 biomedicines-10-02560-f002:**
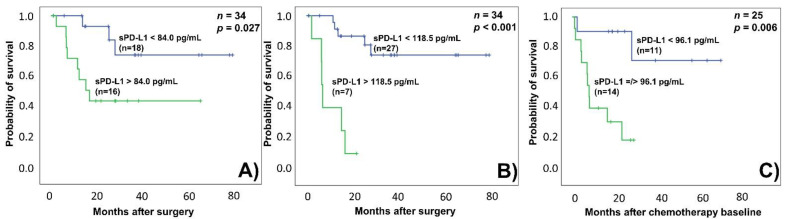
Kaplan–Meier OS analyses with log-rank tests (**A**) for the RNU cohort using the median (84.0 pg/mL) as the cut-off, (**B**) for the RNU cohort applying the ROC-based (118.5 pg/mL) cut-off, (**C**) for the CTX cohort with the median (96.1 pg/mL) cut-off (in this cohort, median- and ROC-based cut-off values resulted the same groups) (blue line—low sPD-L1 cc., green line—high sPD-L1 cc., cut-off values are shown on each line).

**Figure 3 biomedicines-10-02560-f003:**
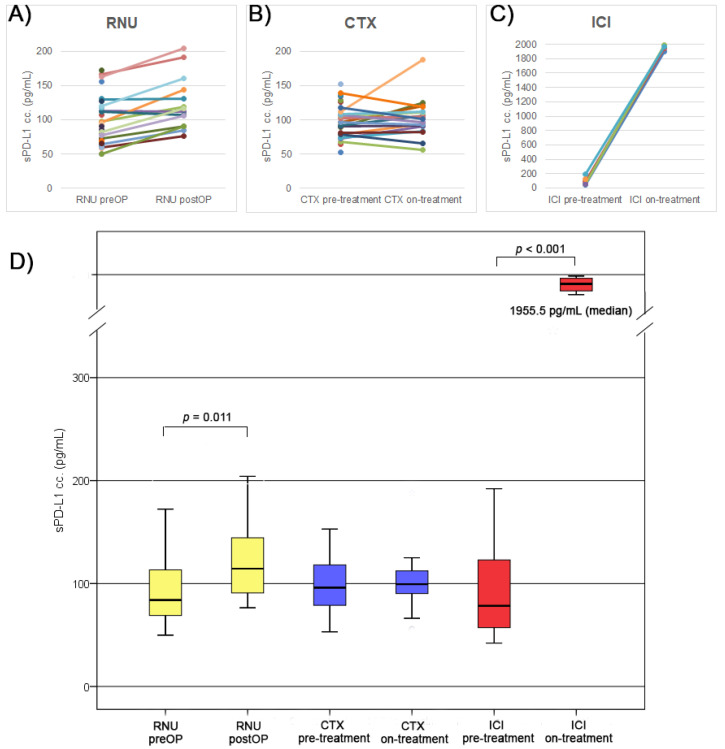
Box-plot presentation of sPD-L1 concentration changes in the RNU, CTX and ICI cohorts. (**A**) RNU cohort (preop. and postop. values), (**B**) CTX cohort (at chemotherapy baseline and on the first day of cycle 2), (**C**) ICI cohort (pretreatment and on-treatment values at 3 months), (**D**). We observed no differences in the pretreatment levels of sPD-L1 between the RNU vs. CTV, RNU vs. ICI and CTX vs. ICI cohorts (*p* = 0.203, *p* = 0.391, *p* = 0.698, respectively). RNU—radical nephroureterectomy; CTX—chemotherapy; ICI—immune checkpoint inhibitor therapy.

**Table 1 biomedicines-10-02560-t001:** Patients’ characteristics for RNU, CTX and ICI treatment groups. * Non-malignant—in three cases of RNU histological examination resulted in a pT0 finding, RNU—radical nephroureterectomy, CTX—chemotherapy, ICI—immun checkpoint inhibitor therapy, ECOG PS—Eastern Cooperative Oncology Group performance status, R+—positive surgical margin, N+—lymph node metastasis, M+—distant metastasis, n. a.—not available, bold font significant value.

	RNU			CTX			ICI	
**General data**	** *n* **	**median (range)**	** *p* **	** *n* **	**median (range)**	** *p* **	** *n* **	**median (range)**
Age at baseline, median (range)	34	68.9 (46.0–90.0)	-	25	72.0 (46.0–84.0)	-	6	64.5 (50.0–76.0)
Follow-up in months, median (range)	34	24.2 (1.1–81.9)	-	25	17.6 (1.1–67.7)	-	6	20.4 (2.6–28.3)
Number of patients died	11	-	-	13	-	-	2	-
**Parameters/sPD-L1 concentrations**	** *n* **	**sPD-L1 cc.**	** *p* **	** *n* **	**sPD-L1 cc.**	** *p* **	** *n* **	**sPD-L1 cc.**
Total No. of patients, median (range)	34	84.0 (49.9–172.3)	0.347	25	96.1 (53.1–152.9)	-	6	78.3 (42.17–192.1)
Non-malignant *	3	68.4 (65.6–83.2)						
Age ≤ 65	10	77.3 (49.9–162.4)	0.183	5	78.6 (53.1–139.5)	0.408	3	94.8 (61.9–122.9)
Age > 65	24	91.4 (59.3–172.3)		20	99.4 (65.0–152.9)		3	57.2 (42.2–192.1)
Sex male	21	93.7 (49.9–172.3)	0.600	21	102.7 (53.1–152.9)	0.452	5	94.8 (57.2–192.1)
female	13	80.7 (57.9–166.1)		4	93.9 (65.0–106.8)		1	42.2
ECOG PS 0	19	80.6 (50.1–166.1)	-	11	89.0 (53.1–128.8)	-	5	61.9 (42.2–192.1)
1	10	89.8 (49.9–162.4)	-	10	103.9 (65.0–139.5)	-	0	-
2	4	98.4 (73.1–172.3)	-	4	107.7 (105.6–152.9)	-	0	-
3	1	119.6	-	0	-	-	1	122.9
ECOG PS 0–1	29	80.7 (49.9–166.1)	0.149	21	91.8 (53.1–139.5)	0.132	5	61.9 (42.2–192.1)
ECOG PS 2–3	5	106.7 (73.1–172.3)		4	107.7 (105.6–152.9)		1	122.9
Localization Ureter	17	70.2 (50.1–166.1)	0.088	13	90.82 (53.1–152.9)	0.298	2	68.5 (42.2–94.8)
Pyelon	10	90.9 (61.7–172.3)		12	106.2 (65.0–139.5)		3	122.9 (61.9–192.1)
Both	7	97.4 (49.9–155.3)		0			1	57.2
**RNU data**								
pT0	3	68.4 (65.6–83.2)	-	-	-	-	-	-
pTa	7	70.2 (50.1–111.7)	-	0	-	-	-	-
CIS	1	57.9	-	0	-	-	1	57.2
pT1	9	68.9 (49.9–113.3)	-	1	135.3	-	1	122.9
pT2	2	110.0 (64.1–155.3)	-	6	80.0 (68.3–128.8)	-	1	94.78
pT3	14	102.0 (72.7–172.3)	-	14	99.4 (53.1–152.9)	-	3	61.9 (42.2–192.1)
pT4	1	126.8	-	2	92.5 (89.02–96.1)	-	0	-
n.a.	0			2			0	
pTa-pT1-CIS (non-invasive)	17	69.4 (49.9–113.3)	**<0.001**	1	135.3	-	2	90 (57.2–122.9)
pT2-pT4 (invasive)	17	106.7 (64.6–172.3)		22	93.9 (53.1–152.9)		4	78.3 (42.2–192.1)
G1	7	62.5 (49.9–85.9)	**-**	0	-	-	0	-
G2	12	87.3 (59.3–117.3)	**-**	5	96.1 (68.3–135.3)	0.951	3	94.8 (61.9–122.9)
G3	15	97.4 (57.9–172.3)	**-**	16	99.4 (53.1–152.9)	-	2	124.6 (57.2–192.1)
n.a.	0			4			1	
G1–G2	19	80.6 (49.9–117.3)	**0.019**		-	-	3	94.8 (61.9–122.9)
G3	15	97.4 (57.9–172.3)			-	-	2	124.6 (57.2–192.1)
R0	26	81.3 (49.9–166.1)	0.368	14	104.2 (65.0–139.5)	0.305	4	76.0 (42.2–192.1)
R+	8	91.9 (64.6–172.3)		9	89.0 (53.1–152.9)		1	61.9
n.a.	0			2			1	122.9
Metastatic status at RNU								
N0/M0	25	76.8 (49.9–155.3)	**0.002**	14	86.0 (53.1–134.5)	0.096	2	76.0 (57.2–94.8)
N+ or M+	9	119.6 (73.1–172.3)		9	102.7 (78.6–152.9)		3	61.9 (42.2–192.1)
n.a.	0			2			1	
Metastatic status at CTX baseline								
M0	-	-	-	10	76.2 (53.1–134.5)	**<0.001**	-	-
M+	-	-	-	14	110.2 (78.6–152.9)		-	-
n.a.	-	-	-	1	-		-	-
CTX regimen								
Gem/Cis	-	-	-	14	89.9 (53.1–125.9)	**0.013**	-	-
Gem/Carbo	-	-	-	11	111.8 (78.6–152.9)		-	-

**Table 2 biomedicines-10-02560-t002:** Characteristics of ICI-treated patients. ICI—immune checkpoint inhibitor therapy, N—lymph node metastasis, M—distant metastasis, Gem/Carb—gemcitabine + carboplatin, Gem/Cis—gemcitabine + cisplatin, n. a.—not available, OS—overall survival, Atezo—atezolizumab, Pembro—pembrolizumab.

	Pat. 1	Pat. 2	Pat. 3	Pat. 4	Pat. 5	Pat. 6
Age	76	64	64	75	65	50
Sex	Female	Male	Male	Male	Male	Male
**Clinicopath. parameters at RNU**						
Stage (pT)	3	2	3	CIS	3	1
Grade (G)	-	2	2	3	3	2
N+	yes	no	yes	no	Yes	no
M+	no	no	no	no	No	unknown
Pre-ICI CTX treatment	Gem/Car	Gem/Cis	Gem/Car	n.a.	Gem/Car	Gem/Cis
**Clinicopath. parameters at ICI baseline**						
ICI-treatment	Atezo	Atezo	Atezo	Atezo	Atezo	Pembro
N+	yes	yes	yes	no	Yes	unknown
M+	yes	yes	no	no	Yes	yes
sPD-L1 at baseline (pg/mL)	42.2	94.8	61.9	57.2	192.1	122.9
sPD-L1 at 3 months (pg/mL)	1903	1939	1993	Unknown	1972	n.a.
OS (months)	14.4	30.2	28.4	28.0	9.9	2.6
status	alive	alive	alive	alive	dead	dead
Objective response	PD	PD	PD	PD	PD	unknown

**Table 3 biomedicines-10-02560-t003:** Correlation of pretreatment sPD-L1 concentrations with patients’ prognosis *—median cut-off value for RNU is 84.0 pg/mL, median cut-off value for CTX is 96.1 pg/mL; **—ROC cut-off value for RNU is 118.5 pg/mL, ROC cut-off value for CTX is 93.9 pg/mL; RNU—radical nephroureterectomy, CTX—chemotherapy; OS—overall survival; PFS—progression-free survival, bold font significant value.

	RNU								CTX					
	OS				PFS				OS			PFS		
	*n*	HR	95% CI	*p*	HR	95% CI	*p*	*n*	HR	95% CI	*p*	HR	95% CI	*p*
Pretreatment sPD-L1														
median cut-off *	17	ref.			ref.			11	ref.			ref.		
median cut-off *	17	4.023	1.060–15.269	**0.041**	2.793	1.011–7.716	**0.048**	14	6.956	1.461–33.110	**0.015**	1.584	0.560–4.478	0.386
ROC cut-off **	27	ref.			ref.			11	ref.			ref.		
ROC cut-off **	7	12.114	2.990–49.082	**<0.001**	6.667	2.140–20.764	**0.001**	14	6.956	1.461–33.110	**0.015**	1.584	0.560–4.478	0.386

## Data Availability

The data that support the findings of this study are available from the corresponding author upon reasonable request.

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
