# Peer review of "High Pretreatment Serum PD-L1 Levels Are Associated with Muscle Invasion and Shorter Survival in Upper Tract Urothelial Carcinoma"

_biomedicines, 2022, doi:10.3390/biomedicines10102560_

Round 1

Reviewer 1 Report

Adam Szeles etal assessed sPD-L1 levels in UTUC patients who underwent radical nephroureterectomy, chemotherapy, or immune checkpoint inhibitor therapy. The results showed n the RNU group elevated preoperative sPD-L1 was as- 32 sociated with higher tumor grade (p=0.019), stage (p<0.001) and the presence of metastasis (p=0.002). 33 High sPD-L1 levels were significantly associated with worse survival in both the RNU and CTX 34 cohorts. sPD-L1 levels were significantly elevated in postoperative samples (p=0.011), while re- 35 maining unchanged during CTX. Interestingly, ICI treatment caused a strong, 25-fold increase in 36 sPD-L1 (p<0.001). This is interesting and important to detect sPD-L1 as a new marker or predictor for tumor stage and worse survival in UTUC. Recommend publishing to Biomedicines after two minor revisions.

1, The author compared two groups using statistic analysis in Figures 1 and 3. Is that possible to use multiple comparison instead two, it will get much more information about the sPD-L1 relationships with different treatments.

2, There is a repeat paragraph at “3.3. Correlation of pretreatment sPD-L1 levels with patients’ prognosis” from Ling 171-178.

3, The author performed an exploratory Spearmen ́s rank correlation analysis between sPD-L1 and MMP-7 in the RNU, CTX and ICI cohorts. The author should show the MMP-7 level in different cohorts.  From the Supplementary Figure 1, it can’t get the information that author showed in the main text: “which revealed a significant positive correlation between the  concentrations of these two serum proteins in the RNU cohort (rs=0.445, p=0.008) but not in the CTX and ICI cohorts (Supplementary Figure 1). “ The author should check and show more information.

Reviewer 2 Report

This study investigated expression of soluble PD-L1 (sPD-L1) in patients with upper tract urothelial carcinoma (UTUC). A concept is interesting, however, there are several unobvious issues in some analyses and lack of clinical meanings.

1.       The results revealed pre and post outcomes of sPD-L1 expression around ICI treatment. How was the association of sPD-L1 expression and ICI objective response rate? If the sPD-L1 merely increased after treating ICI, there could be no clinical meanings. How do you think these incremental results with regard to clinical situation and future prospective?

2.       In addition, there is no statistical analyses between ICI and clinical outcomes like Table 1. But there was statistical significance between pre and on treatment of ICI like Figure 3D. I might think the present study might be selected the better outcomes.

3.       UTUC contained renal pelvic and ureteral cancer. This category needs to be shown in this kind of analysis. In addition, the present study should be demonstrated the difference of clinicopathological outcomes including sPD-L1 data using this category.

4.       There is no data in terms of ROC. This data should be shown because this data is the key of this study.

5.       In Table 3, this table is unobvious. I might think most of the data was described in sentences so that table 3 might be deleted except for “Pretreatment sPD-L1”.

6.       In results 3.5 (line 224), this outcome suddenly showed up. There was no mention in terms of methodology. In addition, this is result section so that there should not quote the references. Even if these results were published before, this study is another concept study so that methodology should be described.

7.       In line 212 (Figure 3C/D, which was…), this sentence was moved in Discussion section.

8.       In discussion line 318-22, there is no citation.

9.       In conclusion, this study just showed the outcomes of sPD-L1 in patents with UTUC, but not in those with bladder cancer. Line 345 should be changed.

Round 2

Reviewer 2 Report

None